# The Value of Ultrasonic Elastography in Detecting Placental Stiffness for the Diagnosis of Preeclampsia: A Meta-Analysis

**DOI:** 10.3390/diagnostics13182894

**Published:** 2023-09-09

**Authors:** Shanshan Su, Yanyan Huang, Weiwen Luo, Shaohui Li

**Affiliations:** 1Department of Ultrasound in Medicine, The Second Affiliated Hospital of Fujian Medical University, Quanzhou 362000, China; susan@fjmu.edu.cn (S.S.); huangyanyan@fjmu.edu.cn (Y.H.); 2Department of Reproductive in Medicine, The Second Affiliated Hospital of Fujian Medical University, Quanzhou 362000, China; 3Department of Ultrasound, Zhangzhou Hospital, Zhangzhou 363000, China; 13959646525@163.com

**Keywords:** elastography, ultrasonography, placental stiffness, non-invasive tool

## Abstract

This meta-analysis evaluated the diagnostic value of ultrasonic elastography in detecting placental stiffness in the diagnosis of preeclampsia (PE). A systematic search was conducted in the EMBASE, Web of Science, Cochrane Library, Scopus database, and PubMed databases to identify studies published before June 2023 using ultrasonic elastography to diagnose PE. The sensitivity, specificity, and diagnostic odds ratio of ultrasonic elastography for diagnosing PE were calculated, and a summary receiver operating characteristic curve model was constructed. The degree of heterogeneity was estimated using the I^2^ statistic, and a meta-regression analysis was performed to explore its sources. A protocol was determined previously (PROSPERO: CRD42023443646). We included 1188 participants from 11 studies, including 190 patients with PE and 998 patients without PE as controls. Overall sensitivity and specificity of ultrasonic elastography in detecting placental stiffness for the diagnosis of PE were 89% (95% CI: 85–93) and 74% (95% CI: 51–89), respectively. The I^2^ values for sensitivity and specificity were 59% (95% CI: 29–89) and 96% (95% CI: 95–98), respectively. The area under the receiver operating characteristic curve was 0.90 (95% CI: 0.87–0.92). The meta-regression analysis showed no significant heterogeneity. Ultrasonic elastography exhibits good diagnostic accuracy for detecting placental stiffness and can serve as a non-invasive tool for differentially diagnosing PE.

## 1. Introduction

The placenta is an important organ for material exchange between the foetus and mother, serving as a life support system for the former. It ensures the efficient exchange of nutrients and gases between the maternal and foetal circulatory systems, supporting the physiological needs of the developing foetus throughout pregnancy [1,2,3,4]. However, defects in the trophoblast invasion of the uterine spiral arteries in a defective placenta can lead to ischaemia-reperfusion injury, resulting in cellular oxidative or endoplasmic reticulum stress. This can lead to the dysfunctional placental secretion of vascular growth factors, including soluble fms-like tyrosine kinase-1 (sFlt-1) and placental growth factor (PlGF), a phenomenon commonly referred to as an angiogenic imbalance [5,6], which can cause maternal vascular inflammatory syndrome, leading to the development of preeclampsia (PE) [5,7]. Early prediction of preeclampsia is necessary to improve maternal and infant outcomes. Screening for preeclampsia based on maternal characteristics and relevant medical history identifies only 35% of patients with PE [8]. Various maternal serum biochemical indices, including activin A, inhibin A, and placental growth factor, have been used to predict preeclampsia; however, the predictive value of these indices is low [9]. Uterine artery Doppler ultrasound has likewise been used to predict PE. Unfortunately, however, most Doppler indices remain poor predictors [10].

Through research on the placenta, we can not only help identify and understand various pregnancy complications, such as gestational hypertension, gestational diabetes, and placental abruption, but also uncover the mechanisms underlying these complications. Medical imaging techniques play a crucial role in helping us understand the effect of placental function on pregnancy-related health and outcomes. Ultrasonic elastography is a non-invasive technique that evaluates the stiffness of target tissues and has been widely used in organs such as the liver, thyroid, and breasts with promising results [11,12,13,14,15,16]. Initial studies on placental elastography measured the elastic modulus of the human placenta during late pregnancy and found that it was independent of the uterine artery pulsatility index, thus serving as an independent assessment parameter [17]. Currently, ultrasonic elastography is widely used to quantitatively evaluate placental stiffness and has been employed as an adjunct diagnostic tool for various perinatal diseases [18,19,20]. Additionally, the stiffness of the placenta is regarded as a potential biomarker for placenta-mediated disease detection [21]. The main elastography techniques used to assess placental stiffness are point shear wave elastography (p-SWE) and 2D shear wave elastography (2D-SWE). The former induces tissue displacement in its normal direction at a single focal point using an acoustic radiation force impulse and measures tissue displacement. Based on this displacement, the lateral shear wave speed will be estimated. The target tissue elasticity would be assessed by the velocity of the lateral propagation of the shear wave [22]. 2D-SWE utilises high-intensity ultrasound pulses to generate shear waves and then measures the velocity of the lateral propagation of the shear wave to assess tissue elasticity. Unlike p-SWE, which focuses on a single focal location, 2D-SWE can interrogate multiple focal zones in rapid succession, allowing real-time monitoring of 2D shear waves, measurement of shear wave velocity or Young’s modulus €, and generation of quantitative elastograms [22,23]. These elastography techniques provide valuable insights into placental health by assessing its stiffness. Changes in stiffness may be associated with pregnancy complications such as placental dysfunction or placental abruption. By using these elastography techniques, healthcare professionals can accurately diagnose and monitor these complications, enabling a more comprehensive understanding of pregnancy health management.

In recent years, there have been an increasing number of studies examining the usage of ultrasonic elastography in diagnosing PE. However, the results have shown significant variability, with sensitivity ranging from 75% to 92%, and specificity ranging from 5% to 92% [24,25,26,27,28,29,30]. Therefore, this meta-analysis aims to provide a comprehensive evaluation of the diagnostic performance of ultrasonic elastography for PE.

## 2. Materials and Methods

### 2.1. The Literature Search Strategy

This study followed the recommendations of PRISMA [31,32]. A protocol was first registered at the PROSPERO registry as CRD42023443646. Systematic searches were conducted in PubMed, EMBASE, Cochrane Library, Scopus database, and Web of Science databases to collect studies published before June 2023 that are related to the usage of ultrasonic elastography in diagnosing PE. The search terms used included PE, preeclampsia, elasticity imaging techniques, elastography, and placenta.

### 2.2. Inclusion and Exclusion Criteria

All articles were independently assessed for inclusion and exclusion by two researchers. The inclusion criteria were as follows: (1) studies that evaluated the accuracy of ultrasound elastography in measuring placental stiffness for diagnosing PE; (2) studies that included 10 or more patients with PE; (3) studies that provided sufficient data to calculate the number of true positive (TP), false positive (FP), false negative (FN), and true negative (TN) cases; (4) studies that assess placental stiffness for diagnosing PE; (5) studies published in English; and (6) original research articles.

### 2.3. Data Extraction

Two investigators independently extracted the data. They extracted and computed the relevant data, including the first author’s name, study country, publication year, patient age, sample size, ultrasound system, ultrasound elastography index, cutoff value, and number of TPs, FPs, FNs, and TNs. If more than one method of diagnosing preeclampsia using placental stiffness was evaluated in a single article, we considered each evaluation as a separate study.

After carefully reviewing the included literature, the two authors used the revised Quality Assessment of Diagnostic Accuracy Studies-2 (QUADAS-2) tool to assess the quality of the studies included in this analysis [33]. The QUADAS-2 tool provides a comprehensive framework to evaluate the risk of bias and methodological quality of each study. However, there may be instances where differences in assessment arise. To resolve these discrepancies, they decided to consult the third author for input and advice. The final decision was negotiated by three authors. This collaborative approach helped to ensure objectivity and consistency in the assessment process, enhancing the reliability of the research quality evaluation. A thorough evaluation of the quality of the included studies is crucial for the effectiveness of the meta-analysis and the accuracy of the conclusions.

### 2.4. Data Analysis

This study employed a random-effects model to calculate the overall sensitivity, specificity, diagnostic odds ratios (DORs), and summary receiver operating characteristic (SROC) curves to evaluate the performance of ultrasound elastography in diagnosing PE. The threshold effect was analysed using the Spearman’s correlation coefficient. Heterogeneity was assessed using the Cochran Q statistic and I^2^ tests. A *p*-value less than 0.05 or I^2^ ≥ 50% indicated substantial heterogeneity among the studies, while a higher *p*-value or I^2^ < 50% indicated minimal heterogeneity. A sensitivity analysis was used to assess the robustness of the results. Potential sources of heterogeneity were explored using a meta-regression analysis. Possible publication bias was assessed using the Deeks’ funnel plot asymmetry test [34]. Statistical significance was set at *p* < 0.05. The meta-analysis was conducted using Stata 15.0 (Stata Corp, California, USA.) and R-language version 4.2.2 (R Foundation for Statistical Computing, Vienna, Austria).

## 3. Results

### 3.1. Literature Search

Figure 1 presents a flowchart of the literature inclusion process used in this study. Initially, 115 studies were identified. After excluding duplicates, 65 remained. Additionally, 58 were excluded, including 6 that had relevant topics but insufficient data. Finally, seven articles were included in the meta-analysis.

This meta-analysis utilised the QUADAS-2 tool to assess the quality of the included studies [33]. Overall, most of the criteria were deemed appropriate and resulted in high QUADAS scores. However, there were shortcomings specifically in the performance bias domain, where only two studies explicitly stated the use of blinding in their research [24,29]. This indicates a suboptimal adherence to blinding protocols among the included studies, potentially introducing bias into the results. It is important to note this limitation when interpreting the findings of this meta-analysis and considering the potential impact on the overall conclusions.

### 3.2. Basic Characteristics of Ultrasound Technology and Literature

This meta-study included a total of nine studies taken from seven articles for evaluation [24,25,26,27,28,29,30]. The basic characteristics of all included studies are presented in Table 1.

Alan et al. [25] analysed the diagnostic performance of PE based on the minimum, average, and maximum shear wave velocities of the placenta. Since Alan et al. [25] had provided sufficient data for all three studies to calculate the number of TP, FP, FN, and TN cases, this meta-analysis decided to treat them as three independent studies for analysis. All studies included only women with singleton pregnancies. Owing to the limited detection depth of the ultrasound probe (typically approximately 8 cm), there were some restrictions in detecting the stiffness of the placenta located in the posterior wall. Therefore, the other studies excluded placentas located in the posterior wall, except for Hefeda et al. [30], who did not specify any restrictions on the placental position. Additionally, except for Sirinoglu et al. [26], who specifically included women in early pregnancy, the other studies focused on mid-term and late-term pregnant women. The characteristics of the ultrasound elastography in the included studies are summarised in Table 2. 

Different ultrasound elastography techniques were used to assess placental stiffness: 2D-SWE in six studies and P-SWE in three studies. Different devices were used for the examination, with two studies using Aiexplorer (SuperSonic Imagine, Provence, France), two studies using ACUSON S2000 (Siemens Medical, Munich, Germany), one study using ACUSON S3000 (Siemens Medical, Munich, Germany), one study using the Samsung HS70A ultrasound system (Samsung healthcare, Gyeonggi-do, Korea), and one study using ElastPQ (Philips Healthcare, Amsterdam, Netherlands). Regarding the choice of measurement units, five studies used shear wave velocity in metres per second (m/s) and four studies recalculated to Young’s modulus using the equation: *E* = 3*ρc*^2^, where *ρ* is the material density and **c** is the wave speed [22]. Additionally, we have provided a summary of the placental stiffness measurement for the different groups of participants in the included studies, as shown in Table 3. 

A quality assessment of the included studies based on the QUADAS-2 tool is provided in the Appendix A.

### 3.3. Diagnostic Performance

This study extracted diagnostic accuracy data from the included studies (Table 4) and analysed them to obtain relevant raw data for the meta-analysis. 

The diagnostic threshold analysis did not reveal heterogeneity, with a Spearman’s correlation coefficient of −0.57 (*p* = 0.112). Therefore, this study evaluated the diagnostic accuracy of placental stiffness for PE by analysing the pooled sensitivity and specificity. The overall sensitivity of the nine trials was 89% (95% CI: 85–93), and the overall specificity was 74% (95% CI: 51–89). The I^2^ values for heterogeneity were 59 (95% CI: 29–89) and 96 (95% CI: 95–98), respectively. The forest plot for evaluating the total sensitivity and specificity of the included studies is shown in Figure 2. The DOR was 24 (95% CI: 8–70). The SROC curve for the combined placental stiffness in diagnosing PE is shown in Figure 3, with an overall area under the SROC curve (AUROC) of 0.90 (95% CI: 0.87–0.92).

### 3.4. Sensitivity Analysis and Meta-Regression Analysis

Sensitivity analysis was performed by synthesizing all studies included in the analysis, and it demonstrated that the results of this meta-analysis were relatively robust (Figure 4). The Cochran Q test and I^2^ test indicated significant heterogeneity in overall sensitivity (*p* = 0.01, I^2^ = 59) and specificity (*p* < 0.001, I^2^ = 96). To further explore the sources of heterogeneity, we conducted a meta-regression analysis. Regression analysis showed that representative values (*p* = 0.76), publication year (*p* = 0.31), total number of cases (*p* = 0.14), and ultrasound elastography techniques (*p* = 0.64) did not contribute to the heterogeneity (Table 5).

### 3.5. Publication Bias Assessment

Deeks’ funnel plot analysis was used to assess publication bias, and no significant asymmetry was found (*p* = 0.81) (Figure 5). 

## 4. Discussion

This meta-analysis included nine studies from seven publications involving 1188 participants, encompassing 190 patients with PE and 998 patients without PE as controls. Ultrasound elastography was found to have had a pooled sensitivity of 89% (95% CI: 85–93%) and a specificity of 74% (95% CI: 51–89%) for diagnosing PE. The pooled DOR was 8 (95% CI: 3–22), and the AUROC curve was 0.90 (95% CI: 0.87–0.92). These results suggest that the use of ultrasound elastography for measuring placental stiffness has a good diagnostic accuracy for PE. Meta-regression analysis indicated no heterogeneity in the representative values (*p* = 0.76), publication year (*p* = 0.31), total number of cases (*p* = 0.14), or ultrasound elastography techniques (*p* = 0.64).

PE is a pregnancy complication associated with a high incidence of morbidity and mortality in mothers and foetuses [35]. The risk of severe complications during the perinatal period is increased 3–25 times in patients with PE, including placental abruption, disseminated intravascular coagulation, pulmonary oedema, and aspiration pneumonia [36]. The pathogenesis of PE is believed to be closely related to placental defects, including vascular growth factors such as sFlt-1 and PlGF [11]. In recent years, some original studies have focused on the value of using ultrasound elastography for evaluating placental stiffness to diagnose PE. However, this situation appears to be complex. Fujita et al. [27] indicated that using p-SWE technology to calculate the mean shear wave velocity (SWV) for evaluating placental stiffness has a sensitivity of 9% and specificity of 91% in diagnosing PE. Moreover, placental stiffness in patients without PE who had high-risk factors was significantly higher than that in patients without PE who did not have high-risk factors, suggesting that placental stiffness increases significantly even before the onset of preeclampsia. Similarly, Alan et al. [25] used p-SWE technology to calculate the mean SWV and reported a sensitivity of 91% and a specificity of 5%. The overall sensitivity and specificity in our study were 89% (95% CI: 85–93%) and 74% (95% CI: 51–89%), respectively, with a DOR of 24 (95% CI: 8–70) and an AUROC of 0.90 (95% CI: 0.87–0.92). These findings suggest that ultrasound elastography, a promising new imaging modality, can help accurately diagnose PE. However, the Cochran Q test and I^2^ test revealed significant heterogeneity in the overall sensitivity (*p* = 0.01, I^2^ = 59) and specificity (*p* < 0.001, I^2^ = 96) in our study. Therefore, we conducted a meta-regression analysis of representative values, publication year, total number of cases, and US elastography techniques. It showed that none of these four subgroups contributed to the heterogeneity, with *p*-values of 0.76, 0.31, 0.14, and 0.64, respectively.

Although all the studies included in this meta-analysis directly or indirectly assessed the stiffness of the target tissue by determining the transverse wave velocity, different devices may employ different algorithms to reconstruct and analyse elastography images of the tissue. These algorithms may have an impact on aspects such as image quality, contrast, and resolution. In addition, different devices have different ultrasound imaging quality and spatial resolution. This will affect the resolving power of elastography and the accuracy of the elastic properties of the tissue. Because the detection depth of most ultrasound probes is only 8 cm, this imposes certain limitations on accurately assessing the stiffness of a placenta located on the posterior wall. Therefore, the accuracy of using elastography to assess the stiffness of a placenta on the posterior wall may be affected. Most of the studies included in this meta-analysis chose to exclude pregnant women with placentas located in the posterior wall of the uterus; however, Hefeda MM et al. [30] did not restrict the location of the placenta, while Sirinoglu HA et al. [26] did not specify whether pregnant women with placentas located in the posterior wall of the uterus were excluded. Additionally, Hall, T.J. et al. [37] indicated that there was a depth-dependent bias in the quantitative values provided by ultrasound elastography. Therefore, the patients included in this meta-analysis may have been confounded by placental positional factors leading to some confounding of the measurement of placental stiffness. Wu et al. [38] demonstrated that there was no significant difference in placental stiffness between mid-term and late-term pregnant women. As a result, most studies included in this meta-analysis enrolled pregnant women from both mid-term and late-term stages. However, the condition of pregnant women with preeclampsia often worsens as gestational weeks progress. Alan et al. [25] showed that the stiffness of the placenta significantly increased as the severity of preeclampsia worsened. Therefore, there may be differences in placental stiffness among pregnant women with preeclampsia at different gestational stages. Edwards et al. [39] found that pre-pregnancy weight status has a significant impact on the detectable stiffness of placental tissue using ultrasound elastography. Placental tissue of obese women exhibited the highest level of stiffness, followed by overweight women. Thus, factors such as limiting the inclusion of participants with the placenta located on the anterior wall or excluding those with the placenta located on the posterior wall, as well as variations in gestational stage and pre-pregnancy weight status among participants, might affect heterogeneity. Unfortunately, due to insufficient reporting or incomplete information in the literature, subgroup analysis to quantitatively investigate these factors could not be performed.

In brief, due to factors such as the inclusion of participants at different gestational stages, restrictions on placental position, and the usage of different types of ultrasound elastography devices and techniques, the recommended optimal cutoff values for PE diagnosis varied among the included studies. In addition, different vendors and probes may yield different stiffness values [22]. Therefore, it is important to recognise the existing differences that can affect the measurement of the target stiffness among the various ultrasound elastography systems and techniques in clinical practice.

To the best of our knowledge, this is the first meta-analysis to evaluate the diagnostic value of ultrasound elastography in detecting placental stiffness in PE. One major strength of this meta-analysis was that it provided a comprehensive assessment of the diagnostic performance of placental stiffness using different ultrasound elastography systems and techniques, including p-SWE and 2D-SWE. Subgroup and meta-regression analyses were performed to explore the sources of heterogeneity.

Our study had certain limitations. First, the number of included studies was relatively small (seven studies). Second, we were unable to access unpublished data, and relying solely on computed data may have affected the reliability of our results. Third, we only included studies published in English, which may have introduced language bias.

## 5. Conclusions

This meta-analysis demonstrates that the use of ultrasound elastography for detecting placental stiffness has a good diagnostic performance for detecting PE. Furthermore, the correlation between ultrasound elastography and corresponding tissue pathological changes warrants further investigation in future studies.

## Figures and Tables

**Figure 1 diagnostics-13-02894-f001:**
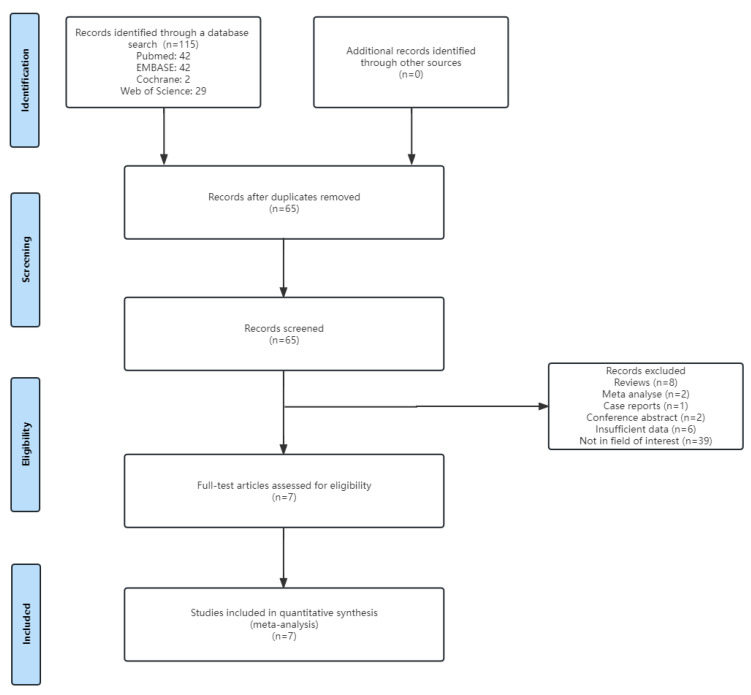
Flow diagram demonstrating the process of selecting eligible studies.

**Figure 2 diagnostics-13-02894-f002:**
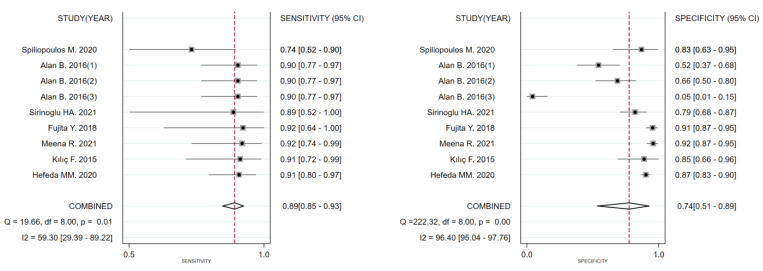
Forest plots of the pooled studies about the sensitivity and specificity of ultrasonic elastography for diagnosing PE [24,25,26,27,28,29,30].

**Figure 3 diagnostics-13-02894-f003:**
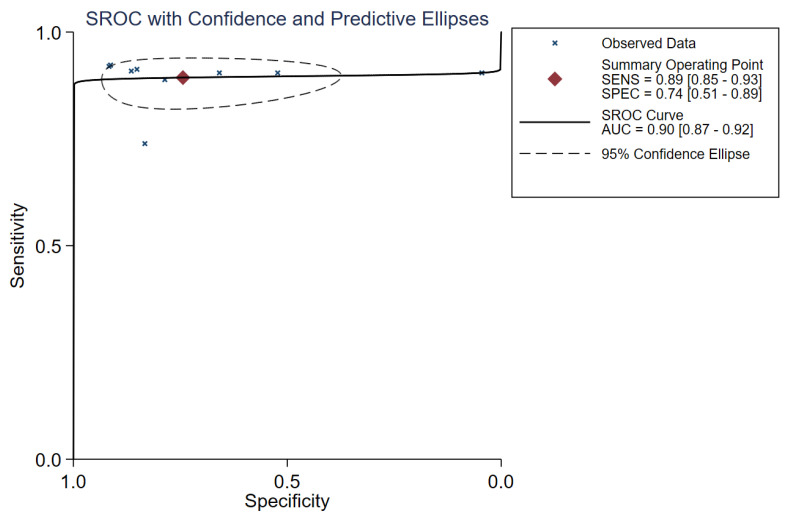
Summary of the receiver operating characteristic (SROC) curve about the diagnostic performance of the pooled studies.

**Figure 4 diagnostics-13-02894-f004:**
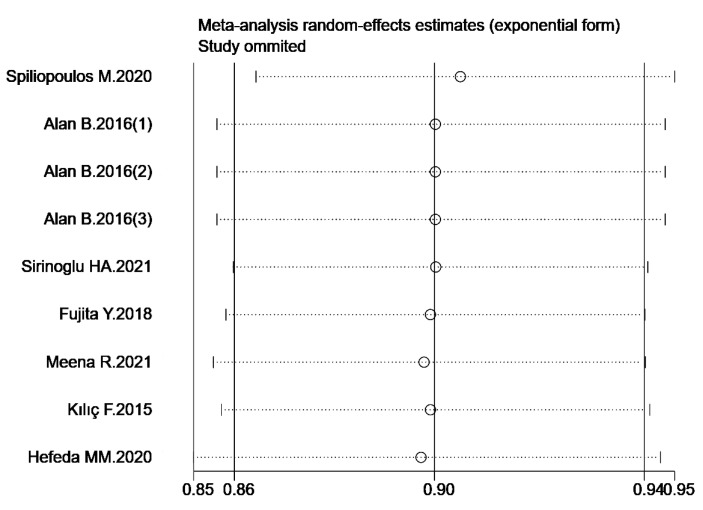
The results of sensitivity analysis [24,25,26,27,28,29,30].

**Figure 5 diagnostics-13-02894-f005:**
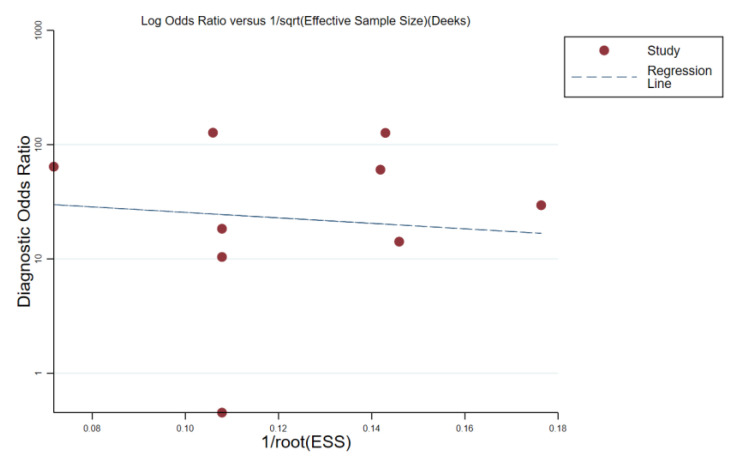
Deeks’ funnel plot for evaluating the publication bias.

**Table 1 diagnostics-13-02894-t001:** Basic Characteristics of the Included Studies.

Number	Author	Year	Country	Study Design	Study Period	Position of Placenta	Trimester of Pregnancy	Group	*n*	Mean/Median Age	Age Range	BMI
1	Spiliopoulos M. [21]	2020	USA	NA	Jan 2017 to Jun 2017	Not posterior placental position	Second and third trimesters	PE group	23	21 ± 2	NA	35 ± 2
								Control group	24	29 ± 1	NA	33 ± 1
2	Alan B. [22]	2016	Turkey	Prospective study	Aug 2014 to Mar 2015	Not posterior placental position	Second and third trimesters	PE group	42	32	28–38	NA
								Control group	44	31	29–37	NA
3	Sirinoglu HA. [23]	2021	Turkey	Prospective study	May 2019 to Dec 2019	NA	First trimester	PE group	9	27 ± 7	NA	26 ± 4
								Control group	75	30 ± 4	NA	25 ± 3
4	Fujita Y. [24]	2018	Japan	NA	Feb 2014 to Mar 2017	Anterior wall	Second and third trimesters	PE group	13	NA	NA	NA
								Control group	208	NA	NA	NA
5	Meena R. [25]	2021	India	Prospective study	Oct 2016 to Jun 2018	Not posterior placental position	Second trimester	PE group	25	24	NA	NA
								Control group	205	25	NA	NA
6	Kılıç F. [26]	2015	Turkey	NA	Jan 2013 to Jun 2013	Not posterior placental position	Second and third trimesters	PE group	23	34 ± 5	22–45	NA
								Control group	27	33. ± 5	26–42	NA
7	Hefeda MM. [27]	2020	Egypt	Prospective study	Oct 2018 to Mar 2020	No restriction	Second trimester	PE group	9	25 ± 7	18–37	NA
								Control group	151	23 ± 6		NA
							Third trimester	PE group	46	27 ± 6	19–42	NA
								Control group	264	24 ± 2		NA

PE: preeclampsia; NA: not available; *n*: number; BMI: body mass index.

**Table 2 diagnostics-13-02894-t002:** Ultrasonic Elastic Technology Characteristics of the Included Studies.

Number	Author	Test Device	Technique	Probe	No. of Measurements	Measurer Blinding	Area of ROI	Depth of Target Tissue
1	Spiliopoulos M. [21]	Aixplorer	2D-SWE	XC6-1 (1–6 MHz)	NA	YES	NA	NA
2	Alan B. [22]	ACUSON S2000	P-SWE	4C1 (2–5 MHz)	4	NA	10 × 6 mm	<8 cm
3	Sirinoglu HA. [23]	Samsung HS70A ultrasound system	2D-SWE	CA1-7A (1–7 MHz)	1	NA	10 × 10 mm	NA
4	Fujita Y. [24]	ACUSON S2000	P-SWE	4C1 (2–5 MHz)	NA	NA	10 × 5 mm	1–6 cm
5	Meena R. [25]	ElastPQ	2D-SWE	ElastPQ (2–5 MHz)	1	NA	NA	<8 cm
6	Kılıç F. [26]	Aixplorer	2D-SWE	XC6-1 (1–6 MHz)	1	YES	5 × 5 mm	NA
7	Hefeda MM. [27]	ACUSON S3000	P-SWE	4C1 (2–5 MHz)	NA	NA	10 × 5 mm	2–8 cm

ROI: region of interest; NA: not available.

**Table 3 diagnostics-13-02894-t003:** Placental Stiffness in Preeclampsia and Control Groups of Included Studies.

Number	Author	Gestational Weeks	Trimester of Pregnancy	Technique	Representative Values	PE Group			Control Group		
						*n*	PSM	Range	*n*	PSM	Range
1	Spiliopoulos M. [21]	>20	Second and third trimesters	2D-SWE	Mean	23	22 ± 3 kPa	NA	24	11 ± 2 kPa	NA
2	Alan B. [22]	27–35	Second and third trimesters	P-SWE	Mean	42	1.4 m/s	1.3–1.5	44	1.1 m/s	1.00–1.1
3	Sirinoglu HA. [23]	23–37	First trimester	2D-SWE	Mean	9	6 ± 2 kPa	2–14	75	9 ± 3 kPa	3–12
4	Fujita Y. [24]	16–32	Second and third trimesters	P-SWE	Median	13	1.4 m/s	1.1–2.4	208	NA	NA
5	Meena R. [25]	16–20	Second trimester	2D-SWE	Mean	25	5 kPa	NA	205	3 kPa	NA
6	Kılıç F. [26]	23–37	Second and third trimesters	2D-SWE	Median	23	21 kPa	2–71	27	4 kPa	2–14
7	Hefeda MM. [27]	>18	Second trimester	P-SWE	Mean	9	2.1 ± 1.5 m/s	NA	46	0.9 ± 0.4 m/s	NA
			Third trimester	P-SWE	Mean	46	2.2 ± 1.5 m/s	NA	94	0.9 ± 0.6 m/s	NA

PE, preeclampsia; PSM, placental stiffness measurement; NA, not available; *n*, number.

**Table 4 diagnostics-13-02894-t004:** Diagnostic Performance of the Included Studies.

Number	Author	Technique Value	Cutoff Value	AUROC	Sensitivity (%)	Specificity (%)	PPV (%)	NPV (%)
1	Spiliopoulos M. [21]	mean SWE value	16 kPa	0.82	75	83	82	76
2	Alan B. [22]	min SWV value	1.0 m/s	0.86	91	47	65	83
		max SWV value	1.7 m/s	0.88	91	67	74	88
		mean SWV value	1.5 m/s	0.99	91	5	50	33
3	Sirinoglu HA. [23]	mean SWE value	7 kPa	0.82	89	79	81	88
4	Fujita Y. [24]	mean SWV value	1.2 m/s	0.91	92	91	40	99
5	Meena R. [25]	mean SWE value	3 kPa	0.97	92	92	58	99
6	Kılıç F. [26]	median SWE value	7 kPa	0.90	90	86	82	92
7	Hefeda MM. [27]	mean SWV value	1.4 m/s	0.91	91	86	73	75

SWE, shear wave elastography; SWV, shear wave velocity; AUROC, area under the receiver operating characteristic curve; PPV, positive predictive value; NPV, negative predictive value.

**Table 5 diagnostics-13-02894-t005:** Results of the meta-regression and subgroup analysis of ultrasound elastography in the diagnosis of PE.

Subgroup		Number of Studies	Pooled Sensitivity (95% CI)	Pooled Specificity (95% CI)	Pooled DOR (95% CI)	Pooled AUROC (95% CI)	*p*-Value
Representative values	Mean	7	0.89 (0.84–0.92)	0.68 (0.39–0.88)	17 (5–59)	0.89 (0.86–0.92)	0.76
	Median	2	0.92 (0.78–0.98)	0.91 (0.86–0.94)	82 (21–323)	NA	
Publication year	≥2020	4	0.88 (0.82–0.92)	0.878 (0.830–0.914)	52 (28–96)	0.94 (0.91–0.96)	0.31
	<2020	5	0.91 (0.85–0.95)	0.60 (0.23–0.89)	15 (3–88)	0.91 (0.88–0.93)	
Total number of cases	≥100	3	0.91 (0.84–0.96)	0.89 (0.87–0.91)	83 (39–177)	NA	0.14
	<100	6	0.88 (0.82–0.92)	0.61 (0.30–0.85)	12 (3–39)	0.88 (0.85–0.91)	
US elastography techniques	2D-SWE	4	0.86 (0.75–0.930)	0.86 (0.78–0.91)	38 (13–110)	0.93 (0.90–0.95)	0.64
	P-SWE	5	0.91 (0.86–0.94)	0.61 (0.23–0.89)	15 (3–87)	0.91 (0.88–0.93)	

DOR: diagnostic odds ratio; AUROC: area under the receiver operating characteristic curve; NA: not available.

## Data Availability

The datasets used and/or analysed in the current study are available from the corresponding author upon reasonable request.

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
