# Peer review of "The Value of Ultrasonic Elastography in Detecting Placental Stiffness for the Diagnosis of Preeclampsia: A Meta-Analysis"

_diagnostics, 2023, doi:10.3390/diagnostics13182894_

Round 1
Reviewer 1 Report
The authors conducted a review about the value of ultrasonic elastography in detecting placental stiffness.
The title of the manuscript is comprehensive.
Overall, the manuscript is well written. English language has good quality.
All keywords were selected from the MeSH (Medical Subject Headings) terminology.
Methods are clear and results are nicely presented.
But the number of included studies is small. Scopus and databases of clinical trials can be included.
In the introduction section, describe the cons and limitations of existing methods for diagnosing preeclampsia.
In the results section mention in which articles more than one method of diagnosing preeclampsia using placental stiffness was evaluated.
You might expand the discussion section by adding information about influence of factors such as the inclusion of participants different gestational stages, restrictions on placental position, and the usage of different types of ultrasound elastography devices and techniques.
Author Response
September 06, 2023
Prof. Dr. Andreas Kjaer
Editor-in-Chief
Diagnostics
Dear Editor:
We wish to re-submit the manuscript titled “The Value of Ultrasonic Elastography in Detecting Placental Stiffness for the Diagnosis of Preeclampsia: A Meta-Analysis”. The manuscript ID is diagnostics-2599854.
We thank you and the reviewers very much for your thoughtful comments and suggestions. The manuscript has benefited from these insightful suggestions. I look forward to working with you and the reviewers to bring this manuscript closer to publication in the Diagnostics.
We have made changes to the original manuscript according to your and the reviewers’ suggestions. All changes are shown in red font in the revised manuscript. The responses to all comments have been prepared and given below.
We hope that our revised manuscript is now acceptable for publication in your journal and look forward to hearing from you soon.
Sincerely.
Shaohui Li
950 Donghai Street
Fengze District, Quanzhou City
Fujian Province, China
Telephone number: +86 136-0073-8911
Email: lishaohui@fjmu.edu.cn
First, we would like to express our sincere gratitude to the reviewers for their constructive and positive comments.
Replies to Reviewer 1
Specific Comments
(1). the number of included studies is small. Scopus and databases of clinical trials can be included.
Response: Thank you for your insightful suggestion. We systematically searched the Scopus database to further increase the number of papers included in this meta-analysis. Unfortunately, the systematic search did not result in an increase in the number of newly-included literature. However, I believe that this initiative increased the scientific validity of this meta-analysis. Please allow me to express my gratitude again for your insight. Lines 85-88 are marked with red font and underlined, and Figure 1 has been updated accordingly.
Systematic searches were conducted in PubMed, EMBASE, Cochrane Library, Scopus database, and Web of Science databases to collect studies published before June 2023 that are related to the usage of ultrasonic elastography in diagnosing PE.
Figure 1. Flow diagram demonstrating the process of selecting eligible studies.
- In the introduction section, describe the cons and limitations of existing methods for diagnosing preeclampsia.
Response: Thank you for your suggestions. Indeed, describing the shortcomings and limitations of existing methods for diagnosing preeclampsia is essential for this meta-analysis, and we strongly agree with your suggestion. Lines 39-45 are marked with red font and underlined.
Early prediction of preeclampsia is necessary to improve maternal and infant out-comes. Screening for preeclampsia based on maternal characteristics and relevant medical history identifies only 35% of patients with PE [8]. Various maternal serum biochemical indices, including activin A, inhibin A, and placental growth factor, have been used to predict preeclampsia; however, the predictive value of these indices is low [9]. Uterine artery Doppler ultrasound has likewise been used to predict PE; unfortunately most Doppler indices remain poor predictors [10].
- In the results section mention in which articles more than one method of diagnosing preeclampsia using placental stiffness was evaluated.
Response: Thanks for your valuable comments. We apologize for the unclear expression that more than one method of diagnosing preeclampsia using placental stiffness was evaluated. We have therefore improved the relevant text. Lines 159-163 are marked with red font and underlined.
Alan et al. [22] analysed the diagnostic performance of PE based on the minimum, average, and maximum shear wave velocities of the placenta. Since Alan et al. [22] had provided sufficient data for all three methods to calculate the number of TP, FP, FN, and TN cases, this meta-analysis decided to treat them as three independent studies for analysis.
- You might expand the discussion section by adding information about influence of factors such as the inclusion of participants different gestational stages, restrictions on placental position, and the usage of different types of ultrasound elastography devices and techniques.
Response: Thank you for raising this critical issue and for your recommendation. We entirely agree and have added information regarding the impact of factors that will make the discussion section more logical. Lines 277-294 are marked with red font and underlined.
Although all studies included in this meta-analysis directly or indirectly assessed the stiffness of the target tissue by determining the transverse wave velocity, different devices may employ different algorithms to reconstruct and analyse elas-tography images of the tissue. These algorithms may have an impact on aspects such as image quality, contrast, and resolution. In addition, different devices have different ultrasound imaging quality and spatial resolution. This will affect the resolving power of elastography and the accuracy of the elastic properties of the tissue. Because the detection depth of most ultrasound probes is only 8 cm, this imposes certain limitations on accurately assessing the stiffness of a placenta located on the posterior wall. Therefore, the accuracy of using elastography to assess the stiffness of a placenta on the posterior wall may be affected. Most of the studies included in this meta-analysis chose to exclude pregnant women with placentas located in the posterior wall of the uterus; however, Hefeda MM et al. [27] did not restrict the location of the placenta, while Sirinoglu HA et al. [23] did not specify whether pregnant women with placentas located in the posterior wall of the uterus were excluded. Additionally, Timothy J. H. et al. [35] indicated that there was a depth-dependent bias in the quantitative values provided by ultrasound elastography. Therefore, the patients included in this meta-analysis may have been confounded by placental positional factors leading to some confounding of the meas-urement of placental stiffness.

Reviewer 2 Report
Dear authors,
The manuscript analyses studies investigating preeclampsia using ultrasound elastography. The review and meta-analysis provide an informative overview of the studies. The meta-analysis combines the findings very well. The manuscript is well-written, but I have a few comments.
1. Line 57: The provided technical description lacks correctness. Both p-SWE and 2D-SWE measure tissue displacement. Based on this displacement, the shear wave speed will be estimated. Some vendors then calculate the Young's modulus based on the measured shear wave speed.
2. Line 58: The phrase "perpendicular to the excitation plane" may be confusing. I assume you mean that they measure the lateral shear wave speed.
3. Inclusion and exclusion criteria. It is unclear at which stage of selecting eligible studies (Fig. 1) the inclusion and exclusion criteria are applied. It is assumed that both inclusion and exclusion criteria are used during the "Identification" stage. If this is the case, what was the purpose of distinguishing between inclusion and exclusion criteria? Would it be possible to combine both criteria to create a total of seven inclusion criteria?
4. Table 1-4. I do not see the benefit of including the year in all of these tables. The study period is the main information and is still given in Table 1.
5. Line 156. Why mention that some studies will be published before or after 2020?
6. Line 169: It should be noted that all studies measure shear wave speed, and four studies recalculate to Young's modulus (which is a more precise term than "elasticity"). Additionally, please include the equation between shear wave speed and Young’s modulus.
7. Line 205. There is observed significant heterogeneity among the studies. It would be helpful to indicate which studies are responsible for this heterogeneity and to consider a subanalysis after excluding these studies.
8. Line 253: It should be noted that the quantitative values provided by ultrasound elastography depend on the depth of the measurement (https://qibawiki.rsna.org/images/f/fe/IEEE2013_QIBA_2013aug9.pdf). Therefore, it is suggested to include some information about the measurement depth, if available in the studies.
9. Line 263-264: The weight affects the measured elastographic values. Therefore, it is suggested to include some information about the body mass index, if available in the studies.
Author Response
September 06, 2023
Prof. Dr. Andreas Kjaer
Editor-in-Chief
Diagnostics
Dear Editor:
We wish to re-submit the manuscript titled “The Value of Ultrasonic Elastography in Detecting Placental Stiffness for the Diagnosis of Preeclampsia: A Meta-Analysis”. The manuscript ID is diagnostics-2599854.
We thank you and the reviewers very much for your thoughtful comments and suggestions. The manuscript has benefited from these insightful suggestions. I look forward to working with you and the reviewers to bring this manuscript closer to publication in the Diagnostics.
We have made changes to the original manuscript according to your and the reviewers’ suggestions. All changes are shown in red font in the revised manuscript. The responses to all comments have been prepared and given below.
We hope that our revised manuscript is now acceptable for publication in your journal and look forward to hearing from you soon.
Sincerely.
Shaohui Li
950 Donghai Street
Fengze District, Quanzhou City
Fujian Province, China
Telephone number: +86 136-0073-8911
Email: lishaohui@fjmu.edu.cn
First, we would like to express our sincere gratitude to the reviewers for their constructive and positive comments.
Replies to Reviewer 2
- Line 57: The provided technical description lacks correctness. Both p-SWE and 2D-SWE measure tissue displacement. Based on this displacement, the shear wave speed will be estimated. Some vendors then calculate the Young's modulus based on the measured shear wave speed.
Response: Thanks for raising this critical issue. We apologize for the error in our technical description here. We have amended it to be accurate. Thank you again for your careful review of our manuscript. Lines 61-65 are marked with red font and underlined.
The former induces tissue displacement in its normal direction at a single focal point using an acoustic radiation force impulse and measures tissue displacement. Based on this displacement, the lateral shear wave speed will be estimated. The target tissue elasticity would be assessed by the velocity of the lateral propagation of the shear wave [19].
- Line 58: The phrase "perpendicular to the excitation plane" may be confusing. I assume you mean that they measure the lateral shear wave speed.
Response: Thank you for your valuable suggestion. We apologize for this error, and we have corrected it. Allow us to reiterate our gratitude for the care and attention you have given to our manuscript. Lines 63-65 are marked with red font and underlined.
The target tissue elasticity would be assessed by the velocity of the lateral propagation of the shear wave [19].
- Inclusion and exclusion criteria. It is unclear at which stage of selecting eligible studies (Fig. 1) the inclusion and exclusion criteria are applied. It is assumed that both inclusion and exclusion criteria are used during the "Identification" stage. If this is the case, what was the purpose of distinguishing between inclusion and exclusion criteria? Would it be possible to combine both criteria to create a total of seven inclusion criteria?
Response: Thanks for your positive comment on the inclusion and exclusion criteria. We also believe that combining the inclusion and exclusion criteria together would be more concise and understandable, as well as more consistent, with the presentation in Figure 1. In addition, the third point of the inclusion criteria was left at one point only because it duplicated the formulation of the third point of the exclusion criteria in terms of information provided by the literature. Lines 93-98 are marked with red font and underlined.
(1) studies that evaluated the accuracy of ultrasound elastography in measuring pla-cental stiffness for diagnosing PE; (2) studies that included 10 or more patients with PE; and (3) studies that provided sufficient data to calculate the number of true positive (TP), false positive (FP), false negative (FN), and true negative (TN) cases; (4) studies that assess placental stiffness for diagnosing PE; (5) studies published in English languages; (6) original research articles.
- Table 1-4. I do not see the benefit of including the year in all of these tables. The study period is the main information and is still given in Table 1.
Response: Thank you for your suggestion. There is really no need for repeated references to the year of publication of the included literature in Tables 1-4, which we have removed in tables other than Table 1.
- Line 156. Why mention that some studies will be published before or after 2020?
Response: Thank you very much for your careful observation. We intended to pave the way for subsequent subgroup analyses of studies published before and after 2020. However, it does seem redundant and illogical. We have removed it from the text, and again, we are grateful for your insight.
- Line 169: It should be noted that all studies measure shear wave speed, and four studies recalculate to Young's modulus (which is a more precise term than "elasticity"). Additionally, please include the equation between shear wave speed and Young’s modulus.
Response: Thank you for your suggestion. We apologize for the lack of rigor in the expression used on this occasion, for which we have corrected the text and added the equation between shear wave speed and Young's modulus to the text. Lines 93-98 are marked with red font and underlined.
Regarding the choice of measurement units, five studies used shear wave velocity in metres per second (m/s) and four studies recalculated to Young's modulus using the equation: , where ρ is the material density and c is the wave speed [19].
- Line 205. There is observed significant heterogeneity among the studies. It would be helpful to indicate which studies are responsible for this heterogeneity and to consider a subanalysis after excluding these studies.
Response: Thank you for your suggestion. We conducted a sensitivity analysis of the included studies based on your suggestion to indicate which studies are responsible for this heterogeneity. Unfortunately, the results of the sensitivity analyses showed that the results of this meta-analysis were relatively robust, and no study was responsible for heterogeneity. Therefore, no studies were excluded from the subsequent subgroup analysis. Lines 221-222 are marked with red font and underlined, and the results of the sensitivity analyses are demonstrated in Figure 4.
Sensitivity analysis was performed by synthesizing all studies included in the analysis, and this demonstrated that the results of this meta-analysis were relatively robust (Figure 4).
- Line 253: It should be noted that the quantitative values provided by ultrasound elastography depend on the depth of the measurement (https://qibawiki.rsna.org/images/f/fe/IEEE2013_QIBA_2013aug9.pdf). Therefore, it is suggested to include some information about the measurement depth, if available in the studies.
Response: Thank you very much for the suggestions. We have provided the information about the measurement depth in Table 2 and added the corresponding information in the Discussion section to increase the scientific quality of the manuscript. Lines 286-294 and Table 2 are marked with red font and underlined.
Most of the studies included in this meta-analysis chose to exclude pregnant women with placentas located in the posterior wall of the uterus; however, Hefeda MM et al. [27] did not restrict the location of the placenta, while Sirinoglu HA et al. [23] did not specify whether pregnant women with placentas located in the posterior wall of the uterus were excluded. Additionally, Timothy J. H. et al. [35] indicated that there was a depth dependent bias in the quantitative values provided by ultrasound elastography. Therefore, the patients included in this meta-analysis may have been confounded by placental positional factors leading to some confounding of the measurement of placental stiffness.
|
Number |
Author |
Test device |
Technique |
Probe |
No. of measurements |
Measurer blinding |
Area of ROI |
Depth of target tissue |
|
1 |
Alan B. |
ACUSON S2000 |
P-SWE |
4C1 (2.0–4.5 MHz) |
4 |
NA |
10 × 6 mm |
< 8 cm |
|
2 |
Kılıç F. |
Aixplorer |
2D-SWE |
XC6-1 (1–6 MHz) |
1 |
YES |
5 × 5 mm |
NA |
|
3 |
Sirinoglu HA. |
Samsung HS70A ultrasound system |
2D-SWE |
CA1-7A |
1 |
NA |
10 × 10 mm |
NA |
|
4 |
Spiliopoulos M. |
Aixplorer |
2D-SWE |
XC6-1 (1–6 MHz) |
NA |
YES |
NA |
NA |
|
5 |
Hefeda MM. |
ACUSON S3000 |
P-SWE |
4C1 (2.0–4.5 MHz) |
NA |
NA |
10 × 5 mm |
1.7 - 7.5 cm |
|
6 |
Meena R. |
ElastPQ |
2D-SWE |
ElastPQ (2–5 MHz) |
1 |
NA |
NA |
< 8 cm |
|
7 |
Fujita Y. |
ACUSON S2000 |
P-SWE |
4C1 (2.0–4.5 MHz) |
NA |
NA |
10 × 5 mm |
1 - 6 cm |
- Line 263-264: The weight affects the measured elastographic values. Therefore, it is suggested to include some information about the body mass index, if available in the studies.
Response: Thank you for your suggestion. We strongly agree that the body mass index (BMI) is important for this meta-analysis. We have therefore added BMI to Table 1.
|
Number |
Author |
Year |
Country |
Study design |
Study period |
Position of placenta |
Trimester of pregnancy |
Group |
N |
Mean/median age |
Age range |
BMI |
|
1 |
Alan B. |
2016 |
Turkey |
Prospective study |
Aug 2014 to Mar 2015 |
Not posterior placental position |
Second and third trimesters |
PE group |
42 |
32 |
28.25–37.50 |
NA |
|
Control group |
44 |
31 |
28.75–37.25 |
NA |
||||||||
|
2 |
Kılıç F. |
2015 |
Turkey |
NA |
Jan 2013 to Jun 2013 |
Not posterior placental position |
Second and third trimesters |
PE group |
23 |
33.5 ±5.3 |
22–45 |
NA |
|
Control group |
27 |
33.3 ±4.69 |
26–42 |
NA |
||||||||
|
3 |
Sirinoglu HA. |
2021 |
Turkey |
Prospective study |
May 2019 to Dec 2019 |
NA |
First trimester |
PE group |
9 |
26.77±7.13 |
NA |
25.61 ±4.22 |
|
Control group |
75 |
29.93±4.29 |
NA |
25.47 ±2.93 |
||||||||
|
4 |
Spiliopoulos M. |
2020 |
U.S.A. |
NA |
Jan 2017 to Jun 2017 |
Not posterior placental position |
Second and third trimesters |
PE group |
23 |
20.9±1.6 |
NA |
34.7 ± 1.6 |
|
Control group |
24 |
28.9±1.4 |
NA |
32.5 ± 1.4 |
||||||||
|
5 |
Hefeda MM. |
2020 |
Egypt |
Prospective study |
Oct 2018 to Mar 2020 |
No restriction |
Second trimester |
PE group |
9 |
24.8±6.5 |
18–37 |
NA |
|
Control group |
151 |
23.4±6.4 |
NA |
|||||||||
|
Third trimester |
PE group |
46 |
27.4±5.6 |
19–42 |
NA |
|||||||
|
Control group |
264 |
24.4±2.3 |
NA |
|||||||||
|
6 |
Meena R. |
2021 |
India |
Prospective study |
Oct 2016 to Jun 2018 |
Not posterior placental position |
Second trimester |
PE group |
25 |
23.92 |
NA |
NA |
|
Control group |
205 |
24.89 |
NA |
NA |
||||||||
|
7 |
Fujita Y. |
2018 |
Japan |
NA |
Feb 2014 to Mar 2017 |
Anterior wall |
Second and third trimesters |
PE group |
13 |
NA |
NA |
NA |
|
Control group |
208 |
NA |
NA |
NA |

Round 2
Reviewer 1 Report
No more comments.
Author Response
We believe that the quality of this manuscript will be greatly improved after this revision. Please allow me to express my gratitude to you for your careful review of our manuscript
Reviewer 2 Report
I thank the authors for a satisfactory revision of the manuscript.
Finally, one more point caught my attention: In some of the figures, the given number of decimal places is too large. E.g. if the age goes from 28.25 - 37.50, it is completely sufficient if 28 - 38 is given. And also the Pooled DOR was given with five significant digits. There 17.407 (5.115 - 59.234) can be reduced to 17 (5 - 59). I ask the authors to check again all (!) given numbers (in the text and in the tables)
Author Response
September 08, 2023
Prof. Dr. Andreas Kjaer
Editor-in-Chief
Diagnostics
Dear Editor:
We wish to re-submit the manuscript titled “The Value of Ultrasonic Elastography in Detecting Placental Stiffness for the Diagnosis of Preeclampsia: A Meta-Analysis”. The manuscript ID is diagnostics-2599854.
We thank you and the reviewers very much for your thoughtful comments and suggestions. The manuscript has benefited from these insightful suggestions. I look forward to working with you and the reviewers to bring this manuscript closer to publication in the Diagnostics.
We have made changes to the original manuscript according to your and the reviewers’ suggestions. All changes are shown in red font in the revised manuscript. The responses to all comments have been prepared and given below.
We hope that our revised manuscript is now acceptable for publication in your journal and look forward to hearing from you soon.
Sincerely.
Shaohui Li
950 Donghai Street
Fengze District, Quanzhou City
Fujian Province, China
Telephone number: +86 136-0073-8911
Email: lishaohui@fjmu.edu.cn
First, we would like to express our sincere gratitude to the reviewers for their constructive and positive comments.
Replies to Reviewer 2
Finally, one more point caught my attention: In some of the figures, the given number of decimal places is too large. E.g. if the age goes from 28.25 - 37.50, it is completely sufficient if 28 - 38 is given. And also the Pooled DOR was given with five significant digits. There 17.407 (5.115 - 59.234) can be reduced to 17 (5 - 59). I ask the authors to check again all (!) given numbers (in the text and in the tables)
Response: Thanks for raising this critical issue. After your suggestion we all agree that too many decimal places will make the expression of the article not clear and concise enough, we have revised the expression according to your requirements for streamlining the processing, expect some of the smaller value of the variables. For example, we have decided to reduce the PSM variate in the table3 to one decimal place to avoid losing too much information. The p-values were reduced to two decimal places to reflect whether the statistical analysis was significantly different. We believe that the quality of this manuscript will be greatly improved with this revision, and please allow me to express my gratitude again for your careful review of our manuscript.
